# Time-restricted feeding has a limited effect on hepatic lipid accumulation, inflammation and fibrosis in a choline-deficient high-fat diet-induced murine NASH model

Tomoyuki Sato [1]*, Katsutaka Oishi [1,2,3]

1 Healthy Food Science Research Group, Cellular and Molecular Biotechnology Research Institute, National Institute of Advanced Industrial Science and Technology, Tsukuba, Japan, 2 Department of Computational Biology and Medical Sciences, Graduate School of Frontier Sciences, The University of Tokyo, Kashiwa, Japan, 3 Department of Applied Biological Science, Graduate School of Science and Technology, Tokyo University of Science, Noda, Japan

* satou.tom@aist.go.jp

**Data Availability Statement:** All relevant data are within the manuscript and its Supporting Information files.

## Abstract

Nonalcoholic steatohepatitis (NASH) occurs worldwide and is characterized by lipid accumulation in hepatocytes, hepatic inflammation, fibrosis, and an increased risk of cirrhosis. Although a major proportion of NASH patients exhibit obesity and insulin resistance, 20% lack a high body mass and are categorized as "non-obese NASH". Time-restricted feeding (TRF), limiting daily food intake within certain hours, improves obesity, lipid metabolism, and liver inflammation. Here, we determined whether TRF affects NASH pathology induced by a choline-deficient high-fat diet (CDAHFD), which does not involve obesity. TRF ameliorated the increase in epididymal white adipose tissue and plasma alanine transaminase and aspartate transaminase levels after 8 weeks of a CDAHFD. Although gene expression of *TNF alpha* in the liver was suppressed by TRF, it did not exhibit a suppressive effect on hepatic lipid accumulation, gene expression of cytokines and macrophage markers (*Mcp1*, *IL1b*, *F4/80*), or fibrosis, as evaluated by Sirius red staining and western blot analysis of alpha-smooth muscle actin. A CDAHFD-induced increase in gene expression related to fibrogenesis (*Collagen 1a1* and *TGFβ*) was neither suppressed by TRF nor that of alpha-smooth muscle actin but was increased by TRF. Our results indicated that TRF has a limited suppressive effect on CDAHFD-induced NASH pathology.

## Introduction

Nonalcoholic fatty liver disease (NAFLD), which has a global prevalence of approximately 25% [1], is defined as hepatic steatosis and injury that occurs without significant alcohol consumption. Nonalcoholic steatohepatitis (NASH) is defined as an advanced stage of NAFLD characterized by hepatocyte injury (ballooning) and inflammation with or without fibrosis [2]. Approximately 20% of NAFLD cases are categorized as NASH [3]. The development of NASH

**Funding:** Japan Society for the Promotion of Science KAKENHI, Grant Number JP21K21273 and JP22K17845 (both to Tomoyuki Sato).

**Competing interests:** The authors have declared that no competing interests exist.

pathology is considered a "multiple-hit" hypothesis, where lipotoxicity to hepatocytes results in the accumulation of oxidative stress that triggers cell death accompanied by inflammation [4]. Chronic inflammation activates hepatic stellate cells (HSCs), and activated HSCs produce collagen and lead to fibrosis [5].

Various rodent models of NASH have been used to clarify the pathology of NASH and identify therapeutic targets. Genetic models include *db/db*, *foz/foz* [6], melanocortin receptor 4 deficient mice [7], and ApoE knockout mice [8, 9]. These models are often combined with obesogenic diets. Obesogenic diet-induced NASH models are produced using diets containing high amounts of fat, cholesterol, and carbohydrates. The drawbacks of these diets are that a long time (12–24 weeks or more) is necessary to develop NASH pathology and they do not result in severe fibrosis [10]. In contrast, diets deficient in methionine and choline exhibit a rapid progression of NASH pathology, with accelerated lipid accumulation in the liver. Mice fed a methionine and choline-deficient diet exhibited hepatic inflammation and fibrosis within 8 weeks [8]. However, this model is not appropriate because it induces significant weight loss rather than obesity. A choline-deficient, L-amino acid-defined high-fat diet (CDAHFD) was developed to induce rapid NASH pathology without significant body weight loss [11]. Male C57BL/6 mice fed CDAHFD exhibited liver fibrosis at 6–10 weeks [11]. Even 1 week of CDAHFD feeding induces liver steatosis, mild inflammation, and fibrogenic responses [12].

A high-fat diet (HFD) disturbs feeding rhythms and induces hyperphagia, mainly during the light phase, leading to obesity [13]. Time-restricted feeding (TRF), which is defined as limited food intake within certain hours of the day, has been reported to improve lipid metabolism. Hatori et al. showed that TRF during the active phase dramatically improved overweight status, liver injury, and hepatic steatosis induced by a HFD [14]. Although TRF suppresses inflammation and obesity [15], its role in liver fibrosis remains unclear. In a clinical study, an 8 h TRF combined with a low-sugar diet for 12 weeks improved liver fibrosis and steatosis scores in patients with NAFLD [16]. Wei et al. revealed that TRF with calorie restriction for 12 months had therapeutic effects on intrahepatic triglyceride content and liver stiffness in patients with NAFLD [17]. However, there is little information on the effects of TRF itself on liver fibrosis, and the underlying mechanism–dependency on the suppression of steatosis and inflammation–remains unclear.

In this study, we evaluated the effects of TRF on the progression of these pathologies using a CDAHFD-induced NASH model, which is generally used to observe severe NASH pathologies, including fibrosis [10]. We found that TRF had limited or no effect on hepatic steatosis, inflammation, and fibrosis induced by CDAHFD.

## Materials and methods

### Animals

Five-week-old male C57BL/6J mice were purchased from Nihon SLC (Hamamatsu, Shizuoka, Japan) and divided into three groups (nine mice/group); mice were housed in groups of three per cage in a 12 h-12 h light-dark cycle (LD 12:12; lights on at Zeitgeber time (ZT) 0 and lights off at ZT12) for 2 weeks. One group was fed CE-2 (CLEA Japan, Inc., Tokyo, Japan) ad libitum, others were fed CDAHFD (A06071302, 60 kcal% fat with 0.1% methionine and no added choline, Research Diets, Inc., New Brunswick, NJ, USA) ad libitum (AL) or under time-imposed restriction where food was accessible to mice during the active phase between ZT14-22 (TRF) for 8 weeks. Body weight and food intake were measured weekly. After 8 weeks, the mice were weighed, and blood was obtained under inhalation anesthesia with isoflurane. After cervical dislocation, liver sections and epididymal white adipose tissue (eWAT) were weighed and rapidly frozen in liquid nitrogen or fixed in 10% neutrally buffered formalin. All animal

experiments were conducted in accordance with the guidelines for animal experiments published by the National Institute of Advanced Industrial Science and Technology (AIST). Our Institutional Animal Care and Use Committee approved all experimental protocols (Permissions: #2023–0398).

## Measurement of blood lipids and parameters

Plasma was obtained by centrifugation at 3000 ×$g$ for 10 min of whole blood collected in EDTA-coated tubes. Alanine aminotransferase (ALT) and aspartate transaminase (AST) activities, triglycerides, non-esterified fatty acids, and total cholesterol in the plasma were measured using kits (Wako Pure Chemical Industries Ltd., Osaka, Japan).

## Hepatic lipid measurement

Total lipids were extracted from the liver sections by Folch's method using chloroform/methanol, with some modifications [18]. After the removal of chloroform by heating at 60°C, the resultant pellet was resolved in isopropanol containing 20% Triton X-100 and preserved at -80°C. Hepatic lipid concentrations were measured using kits (Wako Pure Chemical Industries, Ltd.).

## mRNA extraction and real-time PCR

RNA was extracted from liver sections using RNAiso Plus (Takara Bio Inc., Shiga, Japan) and cDNA was synthesized from 1.0 μg of total RNA using PrimeScript RT reagent kit with gDNA Eraser (Takara Bio Inc.). Real-time PCR was performed using SYBR Premix Ex Taq II (Takara Bio Inc.) and a LightCycler (Roche Diagnostics, Manheim, Germany). The primers used are listed in Table 1.

## Western blotting

Proteins were extracted from frozen liver sections in RIPA buffer containing protease (Roche Diagnostics, Indianapolis, IN, USA) and phosphatase inhibitor tablets (PhosSTOP, Sigma-Aldrich Corp. St. Louis, MO, USA), and quantified using a protein assay kit (Fujifilm Wako Pure Chemicals, Osaka, Japan). Western blotting was performed as previously described [19].

**Table 1. Primer list.**

| TNFα | Fw | TCTCATCAGTTCTATGGCCC |
|---|---|---|
| | Rv | GGGAGTAGACAAGGTACAAC |
| Mcp1 | Fw | TGGAGCATCCACGTGTTGGCTC |
| | Rv | ACACCTGCTGCTGGTGATCCTC |
| IL1β | Fw | CTTCCAGGATGAGGACATGAG |
| | Rv | TAATGGGAACGTCACACACC |
| F4/80 | Fw | CTTTGGCTATGGGCTTCCAGTC |
| | Rv | GCAAGGAGGACAGAGTTTATCGTG |
| TGFβ | Fw | GGATACCAACTATTGCTTCAGCTCC |
| | Rv | AGGCTCCAAATATAGGGGCAGGGTC |
| Col1a1 | Fw | AGGCTTCAGTGGTTTGGATG |
| | Rv | CACCAACAGCACCATCGTTA |
| αSMA | Fw | AGCCATCTTTCATTGGGATGG |
| | Rv | CCCCTGACAGGACGTTGTTA |
| 18S | Fw | GTAACCCGTTGAACCCCATT |
| | Rv | CCATCCAATCGGTAGTAGCG |

Protein (40 μg) was separated by 8% sodium dodecyl-sulfate polyacrylamide gel electrophoresis and transferred to polyvinylidene difluoride membranes (BioRad, Hercules, CA, USA). The membranes were blocked with 4% Block Ace (DS Pharma Biomedical Co., Ltd., Osaka, Japan) for 1 h and probed with the primary antibody (anti-αSMA polyclonal antibody, Proteintech Group Inc., 23081-1-AP, diluted 1:2000) at 4˚C overnight. The proteins were detected using horseradish peroxidase-labeled secondary antibody and an enhanced chemiluminescence system (Wako Pure Chemical Industries). After detection, the membrane was stained with Coomassie Brilliant Blue (CBB) to confirm the equality of the loaded proteins. Proteins were quantified using ImageJ (ver. 1.53k; https://imagej.nih.gov/ij/).

### Sirius red staining

Liver samples fixed with 10% neutrally buffered formalin were embedded in paraffin and sliced into 5 μm-thick sections. Following Sirius red staining, the areas of fibrosis were quantified in 6–10 fields per mouse using ImageJ.

### Statistical analyses

All data are shown as the means ± standard error of the mean. Statistical analysis was conducted using a one-way analysis of variance, followed by Tukey's test, using Microsoft Excel-Tokei 2010 software (Social Survey Research Information Co. Ltd., Osaka, Japan). Differences were considered statistically significant at $P < 0.05$.

## Results

### TRF affected CDAHFD-induced changes in body weight and white adipose tissue weight

We investigated the effects of TRF on the body, liver, and epididymal white adipose tissue (eWAT) weight. As shown in Fig 1A and 1B, TRF decreased calorie intake by 8% and ameliorated the increase in body weight observed in the CDAHFD-AL group. Moreover, CDAHFD tended to increase eWAT weight, and TRF suppressed this increase in eWAT weight (Fig 1C). In contrast, the increase in liver weight induced by CDAHFD was not affected by TRF (Fig 1C).

**TRF showed no effect on steatosis, hepatic inflammation, and fibrosis.** To evaluate the effects of TRF on CDAHFD-induced NASH pathology, we measured plasma liver injury markers, ALT and AST activity, and lipid content in the liver and evaluated inflammatory response and hepatic fibrosis. As shown in Fig 2A, plasma ALT and AST levels were elevated by the CDAHFD; however, ALT levels were lower in the CDAHFD-TRF group than in the CDAHFD-AL group. The absence of a significant difference in AST levels seemed to be caused by the high value of one sample in the TRF group (S1 Fig). Plasma triglyceride levels were similar among the three groups, indicating little difference in lipid excretion from the liver (Fig 2B). In contrast, CDAHFD decreased plasma NEFA levels and increased plasma total cholesterol levels. There were no differences in plasma NEFA and cholesterol levels between the AL and TRF groups (Fig 2B). Hepatic triglyceride levels increased in the CDAHFD-fed mice, and there was no difference between the AL and TRF groups (Fig 3A). In contrast, hepatic cholesterol and NEFA levels were similar among the three groups (Fig 3A). On CDAHFD-induced inflammatory responses, mRNA expression of *Mcp1*, *F4/80*, and *IL1β* were not suppressed by TRF but that of *TNFα* showed a statistically significant decrease in the TRF group compared to the AL group (Fig 3B). Regarding hepatic fibrosis, mRNA expression related to fibrogenesis was elevated in the CDAHFD-fed group and no difference was observed between the AL and

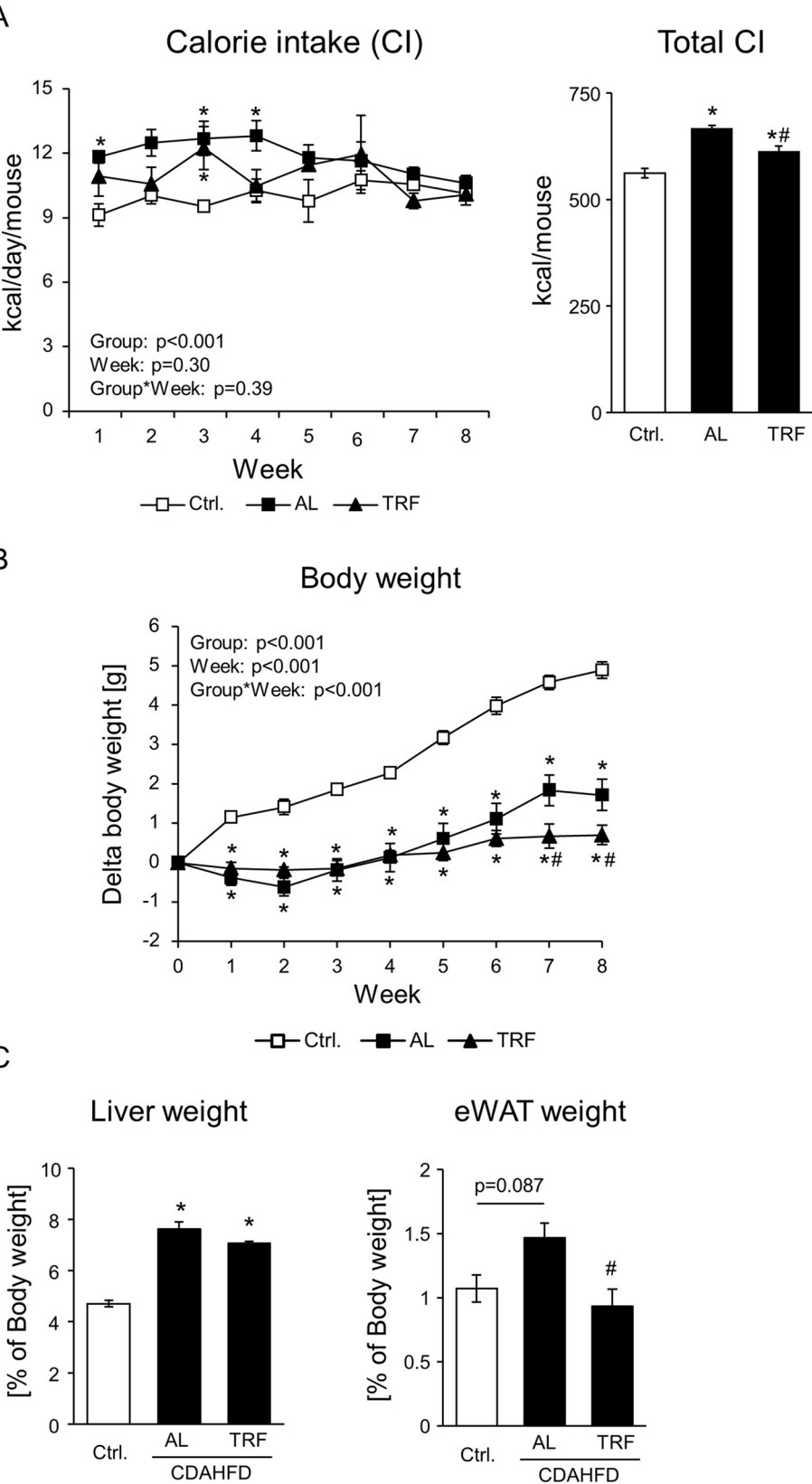

**Fig 1. Effect of TRF on food intake, body weight, and tissue weight in CDAHFD-NASH model.** (A) Daily calorie intake was measured and calculated for every cage. Total calorie intake throughout the experiment is shown on the right. Data are shown as the means ± standard error of means (S.E.; n = 3). *P<0.05 vs Ctrl diet, #P<0.05 vs CDAHFD-AL. (B) Changes in body weight. Data are shown as the means ± S.E. (n = 9). *P<0.05 vs Ctrl, #P<0.05 vs CDAHFD-AL of each timepoint. (C) Liver and eWAT weight at week 8. Data are shown as the means ± S.E. (Liver, n = 8–9; eWAT, n = 5–6). *P<0.05 vs Ctrl, #P<0.05 vs CDAHFD-AL.

TRF groups (Fig 4A), except that *αSMA* expression exhibited further elevation by TRF. CDAHFD elevated protein expression of αSMA in the liver and no effect of TRF was observed (Fig 4B). Similar results were obtained using Sirius red staining (Fig 4C). Thus, most CDAHFD-induced NASH pathologies were not suppressed by TRF.

## Discussion

Time-restricted feeding during the active phase has been reported to improve HFD-induced obesity and liver injury without reducing caloric intake because of the improvement of lipid

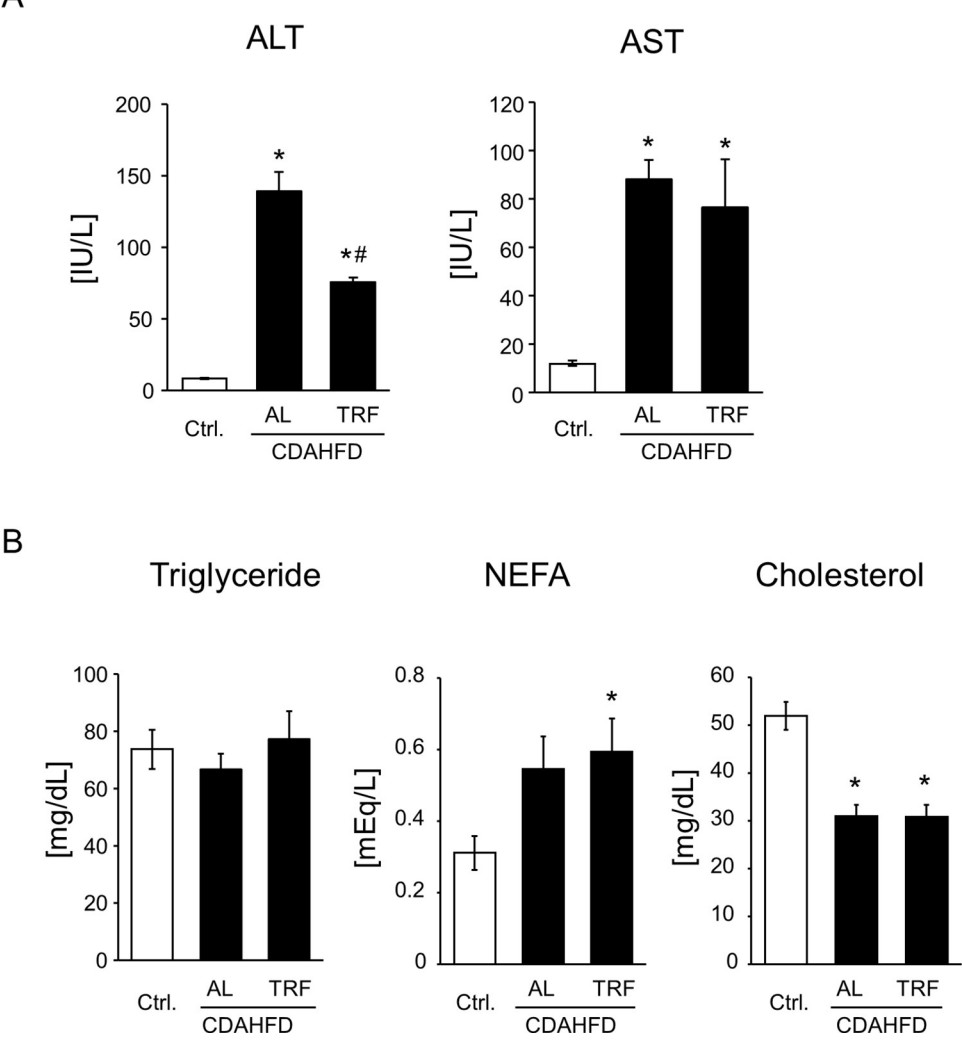

**Fig 2. Effect of TRF on liver injury and plasma lipids.** Liver injury markers (A) and lipids (B) in the plasma. Data are shown as the means ± S.E. (n = 9). *P<0.05 vs Ctrl, #P<0.05 vs CDAHFD-AL.

A

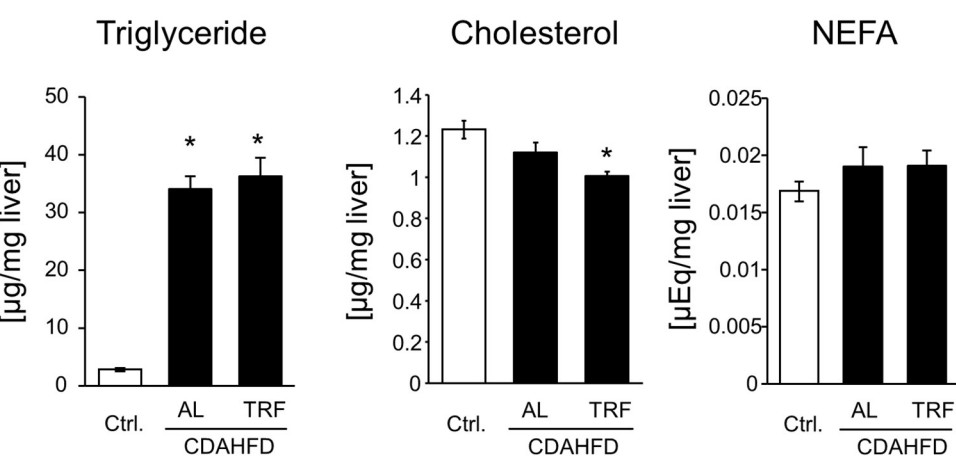

B

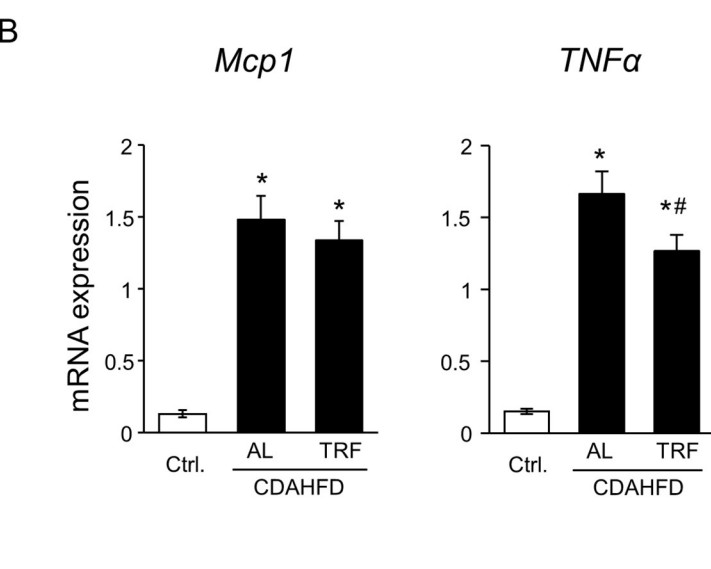

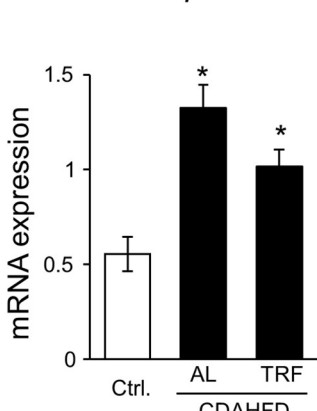

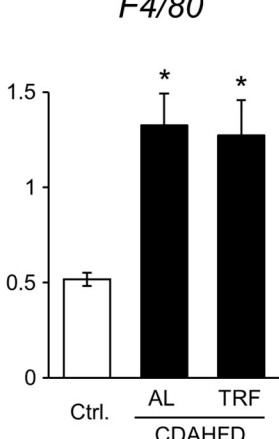

**Fig 3. Effect of TRF on hepatic lipid accumulation and inflammation.** (A) Hepatic lipid content. Triglyceride, total cholesterol, and non-esterified fatty acid were measured. Data are shown as the means ± S.E. (n = 9). *P<0.05 vs Ctrl. (B) mRNA expression of inflammatory markers in the liver. Data are shown as the means ± S.E. (n = 9). *P<0.05 vs Ctrl, #P<0.05 vs CDAHFD-AL.

metabolism, which is mediated by the restoration of feeding rhythm [14]. In this study, TRF decreased total calorie intake by 8% and suppressed the increase in body weight (Fig 1A and 1B). Considering that fasting enhances lipolysis in adipose tissue through activation of adipose triglyceride lipase and hormone-sensitive lipase [20], the lower eWAT weight in the TRF group may be due to sufficient length of the fasting period (16 h a day), in addition to calorie restriction. Moreover, the suppression of body weight gain is consistent with some reports that TRF enhances thermogenesis in adipocytes and energy expenditure [14, 21].

Feeding mice a HFD requires approximately 16–24 weeks to establish NASH pathology. To overcome long-term demands, Matsumoto et al. first reported a CDAHFD as a dietary NASH model [11]. Consistent with their report, 8-week feeding of CDAHFD induced liver injury, hepatic steatosis, inflammation, and fibrosis (Figs 2–4). Time-restricted feeding had limited effects on steatosis, inflammation, and fibrosis (Figs 3A, 3B, and 4). In contrast, TRF suppressed plasma ALT and AST activities, body weight, and eWAT weight (Figs 1B, 1C, 2A, and S1 Fig). These results indicate that TRF does not ameliorate most CDAHFD-induced hepatic pathologies, but suppresses obesity and damage to hepatocytes.

In NASH pathology, hepatic steatosis and hepatocyte death lead to hepatic inflammation via inflammasome activation and release of cytokines [5]. Although it has been reported that TRF suppresses HFD-induced inflammation in the liver [22], it did not suppress the hepatic gene expression of *Mcp1* and *F4/80* (Fig 3B), indicating that the infiltration of macrophages/ Kupffer cells was not ameliorated by TRF. However, a statistically significant decrease in that of *TNFα* by TRF suggests that inflammatory response could be ameliorated, possibly resulting from suppression of hepatocyte injury as observed in plasma ALT and AST (Fig 2A and S1 Fig). No suppression of liver fibrogenesis by TRF (Fig 4A) is conceivable considering the limited suppression of inflammation, as discussed above. Correspondingly, liver fibrosis was not affected by TRF treatment (Fig 4B and 4C). Exacerbated gene expression of *αSMA* (Fig 4A) without change in protein expression in the TRF group (Fig 4B) is possibly caused by the enhanced cycle of activation and resolution of HSCs. Further investigations are needed to understand the effects of TRF on HSCs.

Some clinical studies have revealed that intermittent fasting and TRF have therapeutic effects on NAFLD pathology, especially on obesity, hepatic triglycerides, inflammation, and liver stiffness [16, 23]. TRF also ameliorates systemic inflammatory status [24]. Consistently, TRF has been shown to alleviate murine NASH induced by a Western diet [25]. These reports seem to conflict with our results showing that TRF has a limited effect on steatosis, hepatitis, and fibrosis. This conflict could be caused by differences in the mechanisms underlying hepatic lipid accumulation between NASH models using Western and choline-deficient diets. Western diets induce steatosis, mediating *de novo* lipogenesis in the liver and the accumulation of excess lipids in white adipose tissue. This model exhibits obesity, hyperlipidemia, and insulin resistance [8, 10]. In contrast, as documented above, CDAHFD induces rapid steatosis by inhibiting lipoprotein-mediated lipid export from the liver and therefore does not induce obesity. Our results indicate that although TRF has a suppressive effect on overnutrition-triggered steatosis mediated by the suppression of hepatic *de novo* lipogenesis (as documented in [14]), it does not have an effect on steatosis triggered by the dysfunction of lipid export from the liver, independent of hepatic *de novo* lipogenesis. The difference between clinical studies and our study might be because of the difference in targets; clinical studies were performed on

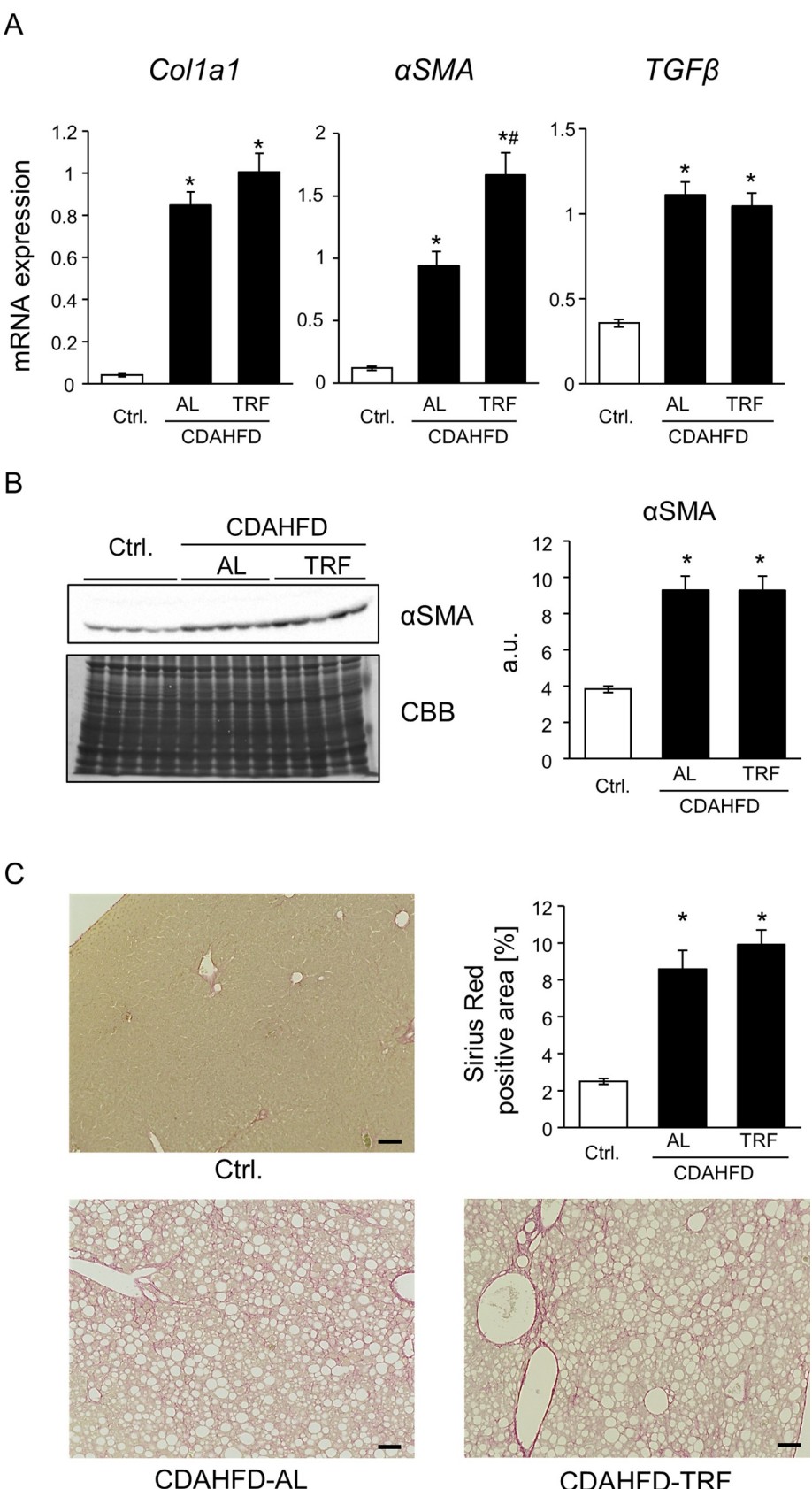

**Fig 4. Effect of TRF on hepatic fibrosis.** (A) mRNA expression of fibrosis markers in the liver. Data are shown as the means ± S.E. (n = 9). *P<0.05 vs Ctrl, #P<0.05 vs CDAHFD-AL. (B) Protein level of αSMA. Data are shown as the means ± S.E. (n = 9). *P<0.05 vs Ctrl. (C) Representative images of Sirius red staining and quantification of Sirius red positive area. Scale bar indicates 100 μm. Averages of Sirius red positive area were calculated for 6–10 images per mouse. Data are shown as the means ± S.E. (n = 9). *P<0.05 vs Ctrl.

obese patients with NASH/NAFLD, setting a lower limit on body mass index or waist circumference [16, 17, 26]. Our study using the NASH model without obesity may indicate that TRF is not therapeutic for patients with NASH/NAFLD without obesity.

In conclusion, TRF did not ameliorate the CDAHFD-induced NASH pathology. However, it is notable that the CDAHFD-induced elevation of plasma ALT and AST was suppressed by TRF. Considering the lack of differences in triglyceride, cholesterol, and NEFA levels in the liver, amelioration of hepatocyte injury might not be caused by improvement of lipid metabolism, but by the protective effect of TRF on hepatocytes. Some reports suggest that TRF or intermittent fasting have protective effects against liver injury by mediating alterations in the gut microbiota [27] or FGF21 [28]. Thus, the protective effects of TRF against lipotoxicity without affecting inflammation and fibrosis in hepatocytes warrants further investigation. As another limitation of this study, the effects of TRF on the composition and subtypes of macrophages/Kupffer cells and dynamics of HSCs in the liver were not investigated. Analysis of these points may provide us with novel insights into NASH pathology.

## Supporting information

**S1 Fig. Dot plot of plasma AST level.** Plasma AST levels were widely distributed in the TRF group. This is mainly owing to the relatively high value indicated by an arrow. The mouse depicted in this figure exhibited mild cirrhosis and a high AST/ALT ratio (2.80), indicating prominent progression of fibrosis [29]. When this sample was eliminated, a statistically significant difference between AL and TRF was detected (P<0.01) using a one-way ANOVA followed by Tukey's multiple comparison test.
(TIF)

**S1 Raw images. Raw data of western blot analysis and CBB staining.** Mice were fed CE-2 diet (Ctrl) or CDAHFD ad libitum (AL) or CDAHFD during active phase between ZT14-22 (TRF) for 8 weeks. Raw images of western blot for αSMA and CBB staining of liver proteins are shown.
(PDF)

## Acknowledgments

We would like to thank Editage (www.editage.jp) for English language editing.

## Author Contributions

**Conceptualization:** Tomoyuki Sato, Katsutaka Oishi.

**Funding acquisition:** Tomoyuki Sato.

**Investigation:** Tomoyuki Sato.

**Methodology:** Tomoyuki Sato, Katsutaka Oishi.

**Project administration:** Tomoyuki Sato.

**Supervision:** Katsutaka Oishi.

**Visualization:** Tomoyuki Sato.

**Writing – original draft:** Tomoyuki Sato.

**Writing – review & editing:** Katsutaka Oishi.

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
