## [Decision Letter · Decision Letter 0]

7 Nov 2023

PONE-D-23-31949Time-restricted feeding has a limited effect on hepatic lipid accumulation, inflammation and fibrosis in choline-deficient high-fat diet-induced murine NASH modelPLOS ONE

Dear Dr. Sato,

Thank you for submitting your manuscript to PLOS ONE. After careful consideration, we feel that it has merit but does not fully meet PLOS ONE’s publication criteria as it currently stands. Therefore, we invite you to submit a revised version of the manuscript that addresses the points raised during the review process.

More specifically, authors should critically revise statements regarding the proposed model as a proxy to human lean NAFLD, as also requested by one reviewer. Moreover, authors report food consumption per mouse: in fact, they never measured this parameter in individual mice, as animals were (presumably) housed in groups (although this is never stated in the MS). Reference to this issue should be made clear in the text.Finally, language should be carefully checked thoughout the MS.

We look forward to receiving your revised manuscript.

Kind regards,

Ezio Laconi, MD, PhD

Academic Editor

PLOS ONE

Journal Requirements:

3. Thank you for stating the following financial disclosure: "Japan Society for the Promotion of Science KAKENHI, Grant Number JP21K21273 and JP22K17845 (both to Tomoyuki Sato)".

Reviewers' comments:

Reviewer's Responses to Questions

**Comments to the Author**

1. Is the manuscript technically sound, and do the data support the conclusions?

Reviewer #1: Yes

Reviewer #2: Partly

2. Has the statistical analysis been performed appropriately and rigorously? 

Reviewer #1: Yes

Reviewer #2: Yes

3. Have the authors made all data underlying the findings in their manuscript fully available?

Reviewer #1: Yes

Reviewer #2: Yes

4. Is the manuscript presented in an intelligible fashion and written in standard English?

Reviewer #1: Yes

Reviewer #2: Yes

5. Review Comments to the Author

Reviewer #1: The manuscript by Sato et al focuses on the effect of time-restricted feeding on the murine NASH model. The topic is state of the art and the manuscript is clearly written for a non-expert. Figures support the text and the results are discussed in relation to the literature. I have no criticism, except that it would be interesting to know the reason for the 12 h restriction (why not e.g. 2*4 h).

Reviewer #2: The authors have performed a study of the effects of 8 weeks of active phase time-restricted feeding (TRF) in a non-obese model of rapidly-induced, advanced NAFLD (CDAHFD). End point analyses included measurements of liver and epidydimal adipose tissue weights, plasma ALT, AST and lipids, liver lipids, liver expression of inflammation and fibrosis genes, and histology for fibrosis. The authors conclude that TRF has limited suppressive effects in non-obese NASH. Given the limited applicability of the model used to human disease, this conclusion is too strong.

1) Statement at the end of the abstract: “Our results indicated that TRF has a limited suppressive effect in non-obese patients with NASH.” This needs to be revised to reflect that the work was done in a mouse model.

2) Was food consumption tracked during the active phase, i.e. was all food that was provided consumed within the ZT14-22 window?

3) Fig. 1: Please provide caloric intake per day, rather than g per day, as it allows for easier determination of whether feeding protocols were isocaloric. Based on the data provided, it appears that there was likely some caloric restriction with this TRF protocol, so this study may be assessing TRF in combination with modest caloric restriction. Would the authors be able to provide AUCs for caloric intake?

4) The authors conclude in the discussion (lines 213-215) that TRF did not decrease total food intake, but suppressed the increase in body weight, therefore indicating that energy expenditure was enhanced by TRF. There is not sufficient evidence provided to support this statement.

5) Please clarify the statement (lines 215-217): “Considering that a HFD narrows the fasting period, suppression of the increase in eWAT weight by TRF could be caused by fasting-triggered lipolysis.”

6) In lines 254-256 the authors state: “Our results indicate that TRF has a suppressive effect on overnutrition-triggered steatosis mediated by the suppression of hepatic de novo lipogenesis (as documented in [14]), but not on steatosis triggered by the dysfunction of lipid export from the liver, independent of hepatic de novo lipogenesis.” In what context does steatosis triggered by the dysfunction of liver lipid export reflect human NAFLD? A lean human would not likely have a defect in lipoprotein secretion, unless in the setting of a genetic mutation. It is difficult to reconcile the CDAHFD model with any form of human NAFLD.

6. PLOS authors have the option to publish the peer review history of their article (what does this mean?). If published, this will include your full peer review and any attached files.

Reviewer #1: No

Reviewer #2: No

---

## [Author Response · Author response to Decision Letter 0]

3 Dec 2023

Thank you for submitting your manuscript to PLOS ONE. After careful consideration, we feel that it has merit but does not fully meet PLOS ONE’s publication criteria as it currently stands. Therefore, we invite you to submit a revised version of the manuscript that addresses the points raised during the review process.

More specifically, authors should critically revise statements regarding the proposed model as a proxy to human lean NAFLD, as also requested by one reviewer. Moreover, authors report food consumption per mouse: in fact, they never measured this parameter in individual mice, as animals were (presumably) housed in groups (although this is never stated in the　MS).　Reference to this issue should be made clear in the text.

Finally, language should be carefully checked throughout the MS.

Response

We deleted statements regarding the CDAHFD model as a lean NASH model from the manuscript and revised the Discussion section as follows: 

(Page 18, line 263-268) The difference between clinical studies and our study might be because of the difference in targets; clinical studies were performed on obese patients with NASH/NAFLD, setting a lower limit on body mass index or waist circumference [16, 17, 26]. Our study using the NASH model without obesity may indicate that TRF is not therapeutic for patients with NASH/NAFLD without obesity.

We also revised the Materials and Method section and figure captions to clarify that food consumption was measured per cage (three mice per cage), not per mouse.

(Page 7, line 85-87) mice were housed in groups of three per cage in a 12 h-12 h light-dark cycle (LD 12:12; lights on at Zeitgeber time (ZT) 0 and lights off at ZT12) for 2 weeks.

(Page 12, line 162) (A) Daily calorie intake was measured and calculated for every cage.

The language of the manuscript has been checked by an English proofreading company, Editage, which has been declared in the Acknowledgments section.

Journal Requirements:

Response

We ensured that our manuscript meets PLOS ONE’s style requirement.

Response

We added statements to the manuscript regarding methods of anesthesia and sacrifice.

(Page 7, line 92-95) After 8 weeks, the mice were weighed, and blood was obtained under inhalation anesthesia with isoflurane. After cervical dislocation, liver sections and epididymal white adipose tissue (eWAT) were weighed and rapidly frozen in liquid nitrogen or fixed in 10% neutrally buffered formalin.

3. Thank you for stating the following financial disclosure: "Japan Society for the Promotion of Science KAKENHI, Grant Number JP21K21273 and JP22K17845 (both to Tomoyuki Sato)".

The funders had no role in this study. We declared this in the cover letter.

Original blot data are attached as “S2 fig” in the Supporting information. In the cover letter, we declared that original blot image data are provided in Supporting information section.

We have revised our manuscript to include captions for the Supporting Information at the end of the manuscript (Page 22, line 377).

We have reviewed and confirmed that the reference list is complete, correct, and does not include any retracted papers.

5. Review Comments to the Author

Reviewer #1: The manuscript by Sato et al focuses on the effect of time-restricted feeding on the murine NASH model. The topic is state of the art and the manuscript is clearly written for a non-expert. Figures support the text and the results are discussed in relation to the literature. I have no criticism, except that it would be interesting to know the reason for the 12 h restriction (why not e.g. 2*4 h).

We aimed to test the effect of time restriction of food intake in the active phase. In this study, we performed 8-h time-restricted feeding according to our previous paper (1). We used this method because recent reports indicate that continuous fasting time is important for anti-obesity effects.

1. Yasumoto Y, Hashimoto C, Nakao R, Yamazaki H, Hiroyama H, Nemoto T, et al. Short-term feeding at the wrong time is sufficient to desynchronize peripheral clocks and induce obesity with hyperphagia, physical inactivity and metabolic disorders in mice. Metabolism. 2016;65(5):714-27.

Reviewer #2: The authors have performed a study of the effects of 8 weeks of active phase time-restricted feeding (TRF) in a non-obese model of rapidly-induced, advanced NAFLD (CDAHFD). End point analyses included measurements of liver and epidydimal adipose tissue weights, plasma ALT, AST and lipids, liver lipids, liver expression of inflammation and fibrosis genes, and histology for fibrosis. The authors conclude that TRF has limited suppressive effects in non-obese NASH. Given the limited applicability of the model used to human disease, this conclusion is too strong.

1) Statement at the end of the abstract: “Our results indicated that TRF has a limited suppressive effect in non-obese patients with NASH.” This needs to be revised to reflect that the work was done in a mouse model.

We revised the abstract to avoid making reference to human NASH as follows:

(Page 3, line 27-29) Here, we determined whether TRF affects NASH pathology induced by a choline-deficient high-fat diet (CDAHFD), which does not involve obesity

(Page 3, line 37-38) Our results indicated that TRF has a limited suppressive effect on CDAHFD-induced NASH pathology.

2) Was food consumption tracked during the active phase, i.e. was all food that was provided consumed within the ZT14-22 window?

Feeding was time-imposed in our experiment; food was accessible ad libitum to mice during ZT14 to ZT22 and was withdrawn from the cages at ZT22. To clarify this point, we revised the Materials and Methods section as follows:

（Page 7, line 88-91） others were fed CDAHFD (A06071302, 60 kcal% fat with 0.1% methionine and no added choline, Research Diets, Inc., New Brunswick, NJ, USA) ad libitum (AL) or under time-imposed restriction where food was accessible to mice during the active phase between ZT14-22 (TRF) for 8 weeks.

3) Fig. 1: Please provide caloric intake per day, rather than g per day, as it allows for easier determination of whether feeding protocols were isocaloric. Based on the data provided, it appears that there was likely some caloric restriction with this TRF protocol, so this study may be assessing TRF in combination with modest caloric restriction. Would the authors be able to provide AUCs for caloric intake?

Calorie intake was calculated and is shown in Fig 1A. Statistical tests revealed that there was a significant difference between AL and TRF in total calorie intake. Therefore, we revised the Results section and Discussion section as follows:

(Page 12, lie 153-154) TRF decreased calorie intake by 8% and ameliorated the increase in body weight observed in the CDAHFD-AL group.

(Page 16, line 218-219) TRF decreased total calorie intake by 8% and suppressed the increase in body weight (Fig 1A, 1B).

4) The authors conclude in the discussion (lines 213-215) that TRF did not decrease total food intake, but suppressed the increase in body weight, therefore indicating that energy expenditure was enhanced by TRF. There is not sufficient evidence provided to support this statement.

Energy expenditure was not measured. We toned down the statements in our conclusions to make it clear that this was merely a hypothesis suggested by some reports but not confirmed by experiments.

(Page 16, line 219-224) Considering that fasting enhances lipolysis in adipose tissue through activation of adipose triglyceride lipase and hormone-sensitive lipase [20], the lower eWAT weight in the TRF group may be due to sufficient length of the fasting period (16 h a day), in addition to calorie restriction. Moreover, the suppression of body weight gain is consistent with some reports that TRF enhances thermogenesis in adipocytes and energy expenditure [14, 21].

5) Please clarify the statement (lines 215-217): “Considering that a HFD narrows the fasting period, suppression of the increase in eWAT weight by TRF could be caused by fasting-triggered lipolysis.”

We revised our manuscript as follows:

(Page 16, line 219-224) Considering that fasting enhances lipolysis in adipose tissue through activation of adipose triglyceride lipase and hormone-sensitive lipase [20], the lower eWAT weight in the TRF group may be due to sufficient length of the fasting period (16 h a day), in addition to calorie restriction. Moreover, the suppression of body weight gain is consistent with some reports that TRF enhances thermogenesis in adipocytes and energy expenditure [14, 21].

6) In lines 254-256 the authors state: “Our results indicate that TRF has a suppressive effect on overnutrition-triggered steatosis mediated by the suppression of hepatic de novo lipogenesis (as documented in [14]), but not on steatosis triggered by the dysfunction of lipid export from the liver, independent of hepatic de novo lipogenesis.” In what context does steatosis triggered by the dysfunction of liver lipid export reflect human NAFLD? A lean human would not likely have a defect in lipoprotein secretion, unless in the setting of a genetic mutation. It is difficult to reconcile the CDAHFD model with any form of human NAFLD.

Response

The statement was based on the fact that choline is necessary to maintain hepatic function and that insufficiency in choline intake such as total parenteral nutrition often leads to fatty liver (1). 

However, the choline deficiency model does not necessarily mimic lean NASH pathology (2). Hence, as the reviewer pointed out, it was inappropriate to consider the CDAHFD model as a human NASH model. We revised our manuscript as follows, simply referring to the difference between NASH induced by CDAHFD and that suffered by patients with obesity.

(Page 18, line 263-268) The difference between clinical studies and our study might be because of the difference in targets; clinical studies were performed on obese patients with NASH/NAFLD, setting a lower limit on body mass index or waist circumference [16, 17, 26]. Our study using the NASH model without obesity may indicate that TRF is not therapeutic for patients with NASH/NAFLD without obesity.

1. Zeisel SH. Choline: an essential nutrient for humans. Nutrition. 2000;16(7-8):669-71.

2. Xu R, Pan J, Zhou W, Ji G, Dang Y. Recent advances in lean NAFLD. Biomed Pharmacother. 2022;153:113331.

---

## [Decision Letter · Decision Letter 1]

21 Dec 2023

Time-restricted feeding has a limited effect on hepatic lipid accumulation, inflammation and fibrosis in choline-deficient high-fat diet-induced murine NASH model

PONE-D-23-31949R1

Dear Dr. Sato,

We’re pleased to inform you that your manuscript has been judged scientifically suitable for publication and will be formally accepted for publication once it meets all outstanding technical requirements.

Kind regards,

Ezio Laconi, MD, PhD

Academic Editor

PLOS ONE

Additional Editor Comments (optional):

Reviewers' comments:

Reviewer's Responses to Questions

**Comments to the Author**

1. If the authors have adequately addressed your comments raised in a previous round of review and you feel that this manuscript is now acceptable for publication, you may indicate that here to bypass the “Comments to the Author” section, enter your conflict of interest statement in the “Confidential to Editor” section, and submit your "Accept" recommendation.

Reviewer #1: (No Response)

Reviewer #2: All comments have been addressed

2. Is the manuscript technically sound, and do the data support the conclusions?

Reviewer #1: Yes

Reviewer #2: Yes

3. Has the statistical analysis been performed appropriately and rigorously? 

Reviewer #1: I Don't Know

Reviewer #2: Yes

4. Have the authors made all data underlying the findings in their manuscript fully available?

Reviewer #1: Yes

Reviewer #2: Yes

5. Is the manuscript presented in an intelligible fashion and written in standard English?

Reviewer #1: Yes

Reviewer #2: Yes

6. Review Comments to the Author

Reviewer #1: The manuscript reviewed, which describes a model of NASH, is state of the art and clearly written. The scientific soundness is improved and the authors answered my question.

Reviewer #2: (No Response)

7. PLOS authors have the option to publish the peer review history of their article (what does this mean?). If published, this will include your full peer review and any attached files.

Reviewer #1: No

Reviewer #2: No

---

## [Editor Report · Acceptance letter]

18 Jan 2024

PONE-D-23-31949R1 

PLOS ONE

Dear Dr. Sato, 

I'm pleased to inform you that your manuscript has been deemed suitable for publication in PLOS ONE. Congratulations! Your manuscript is now being handed over to our production team.

Kind regards, 

on behalf of

Dr. Ezio Laconi 

Academic Editor

PLOS ONE